# A Novel Strategy for Regulating mRNA’s Degradation via Interfering the AUF1’s Binding to mRNA

**DOI:** 10.3390/molecules27103182

**Published:** 2022-05-16

**Authors:** Kun-Tao Li, Xiong-Zhi Wu, Zhi-Yin Sun, Tian-Miao Ou

**Affiliations:** School of Pharmaceutical Sciences, Sun Yat-sen University, Guangzhou 510006, China; likt6@mail2.sysu.edu.cn (K.-T.L.); wuxzh9@mail.sysu.edu.cn (X.-Z.W.); sunzhy0118@163.com (Z.-Y.S.)

**Keywords:** AUF1, RNA-binding protein, mRNA degradation, gene transcription

## Abstract

The study on the mechanism and kinetics of mRNA degradation provides a new vision for chemical intervention on protein expression. The AU enrichment element (ARE) in mRNA 3′-UTR can be recognized and bound by the ARE binding protein (AU-rich Element factor (AUF1) to recruit RNase for degradation. In the present study, we proposed a novel strategy for expression regulation that interferes with the AUF1-RNA binding. A small-molecule compound, JNJ-7706621, was found to bind AUF1 protein and inhibit mRNA degradation by screening the commercial compound library. We discovered that JNJ-7706621 could inhibit the expression of AUF1 targeted gene IL8, an essential pro-inflammatory factor, by interfering with the mRNA homeostatic state. These studies provide innovative drug design strategies to regulate mRNA homeostasis.

## 1. Introduction

RNA-binding proteins (RBPs) are critical trans factors that associate with specific cis-elements in mRNAs, thereby regulating the turnover of target mRNAs by enhancing or repressing mRNA decay [1]. AUF1, also known as hnRNP D (heterogeneous nuclear ribonucleoprotein D), is a ubiquitously expressed RNA binding protein that binds to the target RNAs [2]. According to molecular weight, it has four subtypes, termed p37, p40, p42, and p45 [3]. The p37 subtype has the highest binding affinity with RNA [4]. It can regulate a variety of different physiological pathways, such as inflammation [5], cancer [6], cardiac [7], and aging [8]. Therefore, AUF1 is a crucial regulator of various physiological responses with complex immunological manifestations.

The recognition and binding of AUF1 to its targeted mRNA mainly relies on AU-rich elements (AREs) located in the 3′-untranslated region (3′-UTR) of the mRNAs. AREs are cis-elements in mRNAs and contain tandem repeats of AUUUA or similar sequences [9]. The mRNA stabilizing effects of AUF1 had been investigated by studying the post-transcriptional control of critical inflammatory genes like *bcl-2* [10], *TNFα* [11], and *IL10* [12]. In addition, numerous RNA binding proteins and microRNAs bind to 3′-UTRs to regulate the mRNAs’ stability and translation. One cytokine, IL8, possesses the ARE element in the 3′-UTR and might contribute to the stability of *IL8* mRNA [13]. IL8 has pro-inflammatory effects, and its production supports tumor growth, angiogenesis, and metastasis [14]. The high serum IL8 concentrations could be relevant to some diseases’ progression, such as breast cancer [15], lung cancer [16,17], pancreatic cancer [18], and melanoma [18]. The expression of IL8 can be regulated at different gene expression stages, including transcription, export, post-transcriptional, and translational processes. Among these regulation modes, post-transcriptional regulation plays a vital role in controlling the expression of *IL8* by modulating mRNA stability.

According to the above information, we presented a novel strategy for regulating mRNA’s degradation via interfering with the AUF1’s binding to mRNA. First, we verified the role of AUF1 on tumor cells progression via RNA-immunoprecipitation sequencing (RIP-seq). Based on the finding that AUF1 is mainly involved in the regulation of translation and participated in the regulation of tumorigenesis and development through its targeted genes, we then screened out a possible AUF1 binder from a commercially available library. The hit, a reported CDK inhibitor JNJ-7706621, could bind to the AUF1 and interfere with the interaction between AUF1 and the ARE-containing RNAs. Furthermore, we found that JNJ-7706621 could raise the amount of *IL8* mRNA while inhibiting the expression, and the possible mechanism might be the disruption of the mRNA homeostasis. Based on these findings, we further designed a series of JNJ-7706621 analogs to prove the strategy’s feasibility.

## 2. Results

### 2.1. AUF1 Is Highly Relevant to Tumor Cells Proliferation

There are many kinds of RNA Binding Proteins (RBP) in eukaryotes, and playing essential roles in many post-transcriptional processes, such as RNA shearing, transport, alternative splicing, intracellular localization, and translation initiation. AUF1 is an RBP that binds to its targeted mRNA through AU-rich elements (AREs) to promote mRNA decay [2]. To verify the relationship between AUF1 and tumor cells proliferation, we first used RNA Immunoprecipitation sequencing (RIP-seq) with an AUF1 antibody to detect the function of AUF1 in Hela cells.

RIP kit was used to extract RNA-protein complexes from Hela cells as samples, and the Input group was the total RNAs library. Statistics of all annotation results by RIPSeeker could reflect the nuclear localization of AUF1. As shown in Figure 1a, 50% of the binding region of AUF1 was in the exon region, and others were mainly in the 3′-UTR regions (29.6%). The high percentage of immunoprecipitated regions in 3′-UTR was consistent with AUF1’s mRNA decay function. We then analyzed the Gene Ontology (GO) of AUF1’s targeted genes and obtained the distribution of candidate genes (Appendix A). The candidate genes were cataloged as cellular-processes-related, cellular-components-related, and binding genes. Most of the targeted genes were identified as protein binding genes in the biological functions class. Cellular-components-related genes accounted for a more significant proportion, which might come from the predominant existence of this kind of gene. So, we focused on cellular-processes-related genes and found that AUF1’s targeted genes were related to metabolic and single-organism processes.

We performed a further biological pathway analysis based on the Kyoto Encyclopedia of Genes and Genomes (KEGG) database. AUF1’s targeted genes were mainly concentrated in the protein synthesis process of the endoplasmic reticulum and cancer pathway (Figure 1b), indicating that AUF1 might be involved in the regulation of translation and participated in the regulation of tumorigenesis and development through its target genes.

### 2.2. Screening Out and the Verification of AUF1’s Binder

At the first beginning of finding an interrupter of AUF1, we applied the surface plasmon resonance (SPR) assay to screen candidate small molecules that bind to AUF1. SPR measured the change in the refractive index of a metal surface and could monitor the binding of molecules to protein immobilized on the metal surface [19,20].

To identify AUF1-interacting compounds, we screened 1890 compounds in the commercial drug library (MedChemExpress, China). The libraries contained approved kinase inhibitors, anti-inflammatory drugs, and antibacterial drugs that target ion channels and signaling pathways. We set the screening concentration of compounds at 50 μM and immobilized purified AUF1 protein on a CM5 chip. According to the coupling number of AUF1 on the chip, we set the RU range from 20 to 120. Thus, we selected 186 compounds from the criterion (shown in the red area in Figure 2a, and the RU values were shown in Appendix A). These compounds were further applied to the kinetic analysis.

However, only one compound, JNJ-7706621, showed a typical binding curve shape (Figure 2b), rapidly reaching a steady-state upon injection and then rapidly dissociating from the binding site. The curve indicated a dynamic behavior of “fast-on and fast-off”. Besides, the highest concentration of JNJ-7706621 could make the highest RU reach the expected highest RU value (according to the immobilizing amount), suggesting the protein was bound to the compound in a 1:1 manner.

Binding affinity is just one aspect of identifying an active molecule. Compound binding often affects the thermal stability of proteins, and the shift in the melt curve indicates cellular target engagement [21]. To verify whether JNJ-7706621 could bind AUF1 in cells, we performed a cellular thermal shift assay (CETSA). CETSA is a label-free method that assesses the thermal stability of proteins in living cells and tissue based on denaturation and aggregation because of heating [22]. Three different cells were selected in this assay: human hepatocellular carcinoma cells HepG2, human skin malignant melanoma cells SK-MEL-2, and human colon cancer cells HCT-116. The cells were treated with JNJ-7706621 at 6, 3.5, and 1 μM, respectively, for one hour. Specifically, the different concentrations used in these three cells depended on the IC_50_ values in the MTT assay. Western blot was used to detect the remaining AUF1 level. We observed that the protein reduced with increasing temperature in three cells and disappeared at a specific temperature (Figure 2c). The trends of disappearing in JNJ-7706621-treated cells differed from those in control cells. We then performed the gray analysis of bands by Image J and established the melting curve (Figure 2d). The melting temperature fitted from the curve in each cell decreased when treated with JNJ-7706621, indicating that JNJ-7706621 might interact with AUF1 and make it more unstable in vivo. Since the conformational change would change the thermal stability, we hypothesized that the JNJ-7706621 treatment might influence the conformation of AUF1.

### 2.3. RNA-seq Analysis for Verifying the Effect of JNJ-7706621

RNA sequence technology can detect RNA expression at a specific time point through high-throughput sequencing and quantitative methods, studying and analyzing transcriptomes among different samples. Since we identified that JNJ-7706621 could interact with AUF1 in cells, we wanted to explore how it affected the cellular processes and the possible relationship between AUF1 functions. To do that, SK-MEL-2 cells were treated with JNJ-7706621 at 1.75 μM for 48 h and analyzed the genetic difference between the vehicle-control and treated groups. As shown in Figure 3a,b, the up-regulated genes in differentially expressed genes (DEG) accounted for 70% of the total genes, indicating that JNJ-7706621 enhanced the number of mRNAs. Through gene ontology (GO) analysis of up-regulated genes, we found that these genes were mainly related to the cellular response to lipopolysaccharide, tumor necrosis factor, biotic stimulus, or interleukin-1 (Appendix A). Furthermore, the KEGG analysis of up-regulated genes indicated that those genes mainly concentrated in cytokine-cytokine receptor interaction and the TNF signaling pathway (Figure 3c). All the sequencing data indicated the potential of JNJ-7706621 in tumor therapy and immune response.

### 2.4. JNJ-7706621 Regulated IL8 mRNA Level via Disturbing RNA-AUF1 Complex Formation

Combining data from the RNA-seq and RIP-seq (Appendix A), we selected three genes that are upregulated by JNJ-7706621 and bound to AUF1 as well. They were *IL8*, *BTG2*, and *PLEK2* (Figure 4a). We did the overexpression and knock-down assays for AUF1 to further confirm whether these genes were the target of AUF1. Since the primary function of AUF1 was to initiate mRNA degradation, we evaluated the remaining mRNAs after AUF1 overexpression and knock-down. As shown in Figure 4b, all the genes exhibited an increase in mRNA amount after being treated with AUF1’s siRNA. At the same time, we transfected the AUF1 expression plasmid into SK-MEL-2 cells and found *IL8* and *PLEK2* mRNA reduced dramatically (Figure 4c). Comprehensively considering all the above data, we picked *IL8* as the AUF1 target gene for further studies since it exhibited the most significant change by AUF1 and its role in immune response and cancer change.

We then detected the mRNA level of *IL8* in SK-MEL-2 cells after treating with JNJ-770621. As shown in Figure 4d, *IL8* mRNA significantly increased, and AUF1 mRNA remained unchanged after treatment. This result was consistent with RNA-seq that JNJ-7706621 increased most mRNA’s amount.

IL8 is a pro-inflammatory factor with a short half-life to rapidly respond to changes in the surrounding environment. As mentioned above, the binding of AUF1 to the ARE motif in 3′-UTR could recruit RNase to degrade surplus mRNA in cells. Therefore, we detected the remaining *IL8* mRNA after treating with JNJ-7706621. Using Actinomycin D to stop the transcription at specific time points, we evaluated the mRNA amount by qRT-PCR. The fitting curve was performed using GraphPad Prism (Figure 5a). The mRNA amount decreased with time in the control group while increased in the JNJ-7706621 treated group. Compared with the control group, mRNA levels were higher at each time point after treated with JNJ-7706621, indicating an inhibition effect on mRNA decay.

Subsequently, we identified whether JNJ-7706621 could block the binding of AUF1 to the ARE motif. Using an ARE model sequence from TNFα, we performed the RNA electrophoresis mobility shift assays (REMSA) and found that AUF1 could bind to this sequence in a dose-dependent manner (Appendix A). After adding increasing JNJ-7706621, the bands representing RNA-AUF1 complex weakened, and the bands of RNA-free strengthened (Figure 5b). These results suggested that the binding of JNJ-7706621 to AUF1 prevented the RNA-protein complex formation, thus inhibiting AUF1’s function in mRNA decay.

To further confirm whether the effect of JNJ-7706621 on the AUF1-RNA complex relied on the ARE element of IL8, we applied a dual-luciferase assay using reporter plasmids containing a wide-type 3′-UTR or a deletion of the whole ARE sequence in 3′-UTR (Appendix A). As shown in Figure 5c, the deletion of ARE element would cause mRNA increase, which was consistent with the reported function of AUF1. JNJ-7706621 exhibited a similar effect of ARE’s deletion, showing increased relative luciferase activity on wild-type reporter plasmid (Figure 5d).

Interestingly, we then evaluated the effect of JNJ-7706621 on IL8 expression, while found the expression decreased instead of increasing with the mRNA accumulation (Figure 5e). To investigate whether AUF1 is involved in IL-8 expression in cells treated with JNJ-7706621, we performed AUF1 knock-down experiments. When AUF1 was knocked down, the expression of IL8 was slightly down-regulated (Figure 5f). At the same time, the expression of IL8 was down-regulated when JNJ-7706621 was added. This result suggested that JNJ-7706621 reduced the expression of IL8 protein partially via AUF1 (but there might exist some other mechanism). The possible reason for this result might come from a negative feedback regulation mechanism in vivo when mRNA played an excessive accumulation or another *cis* factors. As a result of the inhibition on IL8 expression, JNJ-7706621 exhibited inhibitory activities on the proliferation of A549, Hela, MCF-7, HepG2, and SK-MEL-2 cells (Figure 5g).

### 2.5. Novel JNJ-7706621′s Analogs Showed Similar Inhibitory Effects on RNA-AUF1 Complex

The findings on JNJ-7706621 provided a good tool for verifying that the strategy interfering with AUF1-RNA binding could break the homeostatic mRNA decay and thus regulate IL8’s expression. However, we must admit that it is hard to distinguish whether this inhibition comes from the interaction with AUF1 or other targets since JNJ-7706621 was first developed as an Aurora kinase inhibitor [23]. To conquer that, we needed to develop novel and specific AUF1 binders based on JNJ-7706621.

The RNA recognition motifs (RRM) conformation of AUF1 had been reported [24], showing that the conformations of the β2-β3 (residues 112–121) loops were intrinsically flexible and involved in the specific recognition of RRM-containing proteins. To investigate JNJ-7706621 interaction mode with AUF1 (PDB code: 5IM0), molecular docking was performed. As illustrated in Figure 6a, JNJ-7706621 showed hydrophobic interactions with LEU109, the sulfamide group formed H-bonds with LEU85 and GLY120, and the triazole group formed H-bond with SER118.

We designed three analogs containing the 1,2,4-triazole scaffold of JNJ-7706621 in the proof-of-concept step (Figure 6b). The synthesis of these compounds used methods developed by Webb [25] (Appendix A). We evaluated the three 1,2,4-triazole-3,5-diamine derivatives using the REMSA assay. As shown in Figure 6c–e, compounds **1** and **3** could interfere with the formation of the AUF1-RNA complex, while compound **2** could not show any significant effect on the complex. Comparing the structures, the replacement of the trifluoromethyl group seemed to work. However, a credible structure-activity relationship (SAR) could not be done until we get enough compounds with diverse variations, which is under-going for now.

## 3. Discussion

The study on the mechanism and kinetics of mRNA’s degradation provides a new vision for chemical intervention on protein expression. The levels of cellular mRNA transcripts can be regulated by controlling the rate at which the mRNA decays. Because decay rates affect the expression of specific genes, they provide flexibility in effecting rapid change when cells respond to stimulus. Moreover, many clinically relevant mRNAs—including several encoding cytokines, growth factors, and proto-oncogenes—are regulated by differential RNA stability [26]. The AU enrichment element (ARE) in the 3′-UTR regions can be recognized and bound by the ARE binding protein (AU-rich Element factor (AUF1) to recruit RNase for degradation [2,27,28]. ARE-containing mRNAs encode many essential proteins in inflammation and cancer and conservatively account for about 10–15% of all transcripts [29]. Our RIP-seq data reinforced the essential role of AUF1 in RNA binding and inflammation or tumor-related pathways. So intervening in the recognition between AUF1 and ARE in mRNAs seemed to be an attractive strategy for finding novel anti-cancer drugs or anti-inflammation drugs.

To find a small molecule that could interfere with the AUF1-RNA interaction, we started with screening AUF1 binders via SPR technology. The library is commercially available (MedChemExpress) and contains 1890 approved drugs or peptidomimetics for their potential in drug safety or mimics the binding domain or proteins. Unfortunately, only one compound could meet the criterion of specific binding, which was JNJ-7706621. From that point, whether SPR screening technology is suitable for RNA-binding protein remains unclear. Thus, we focused on the activity of interfering with the interaction of AUF1 and ARE-containing RNAs and found that JNJ-7706621 could block the complex forming in vitro. Since RNA-binding protein possesses an ‘undruggable’ pocket and mostly experiences an allosteric process when it binds to RNA, more screening methods must be taken into consideration.

To further verify the effect of JNJ-770621, we applied several in vitro and cellular assays, including EMSA, qRT-PCR, CETSA, overexpression and knock-out, dual-luciferase assay, RNA sequencing, and remaining mRNA evaluation. JNJ-770621 could not only interact with AUF1 but also interfere with the function of AUF1 through the block on AUF1’s binding to mRNA. Interestingly, JNJ-7706621 showed the opposite effect on transcripts (increase) and protein expression (decrease). The possible mechanism is that breaking down the homeostatic state of mRNA decay causes surplus mRNA accumulation in cells, but we still need more evidence. In addition, IL8 expression is regulated by several distinct mechanisms through the 3′-UTR since there exist multiple binding sites of trans-acting factors, such as miR-93-5p [30], miR-520c-3p [31], HuR [32], KSRP [13]. Therefore, the possibility that JNJ-7706621 administration prevented the interaction between AUF1 and IL8 mRNA but activated other pathways to inhibit IL8 expression [33,34,35].

Although JNJ-770621 showed significant inhibitory activity on tumor cell proliferation, it is hard to tell the effect of its inhibitory action on AUF1. Since we aimed to find a molecular tool at the beginning of when we proposed the novel strategy, we did not do lots of evaluation experiments to distinguish the significant reason that JNJ-7706621 exhibited anti-tumor activity. Alternatively, we thought finding a molecule with a more specific effect on the AUF1-RNA complex was necessary for further development. According to the proof-of-concept experiments, designing JNJ-7706621 analogs seemed to be an effective strategy. We are now extending the structural variety of 1,2,4-triazole derivatives (undergoing work, data now shown).

## 4. Materials and Methods

### 4.1. Surface Plasmon Resonance (SPR) Assay

The SPR experiments were performed on a Biacore 8K instrument (GE Healthcare) equipped with a CM5 sensor chip. The sensor chip was activated via an amine-coupling reaction with EDC and NHS. Then, p37^AUF1^ was applied to a flow cell with 20 mM PB buffer at pH 7.4. p37^AUF1^ was immobilized at a density of 10,000 RU (response units). The surface was blocked with 1.0 M ethanolamine hydrochloride at pH 8.5. The affinity analysis was carried out for 1980 compounds, and the initial screening concentration was 50 μM. By comparing all aspects of consideration, we screened out 186 compounds for kinetic analysis. The screening concentration of 186 compounds was roughly set as 80, 20, 5, and 0 μM and made refined *K*_D_. Since the final concentration of DMSO in the sample was 5% (*v*/*v*), we set the solvent correction range of 4.7–5.3%.

### 4.2. RNA Electrophoresis Mobility Shift Assay (REMSA)

The REMSA used recombinant p37^AUF1^ and FAM-labeled TNF ARE oligomer (5′-GUGAUUAUUUAUUAUUUA UUUAUUAUUUAUUUAUUUAG-3′) [30]. The buffer used in the whole experiment was 10 mM Tris-HCl (pH 8.0) containing 50 mM KCl, 2 mM DTT, 0.5 mM EDTA, and 0.1 µg/µL BSA. RNA oligomer was heated at 95 °C for 5 min and then annealed by graduate cooling to room temperature. 250 nM FAM-RNA and different concentrations of p37^AUF1^ were co-incubated on ice for 10 min in a final volume of 20 µL. The reaction mix was then loaded onto a 6% native polyacrylamide gel in 1× TBE buffer at 150 V and 4 °C for 50 min. When compounds were added in the reaction mix, compounds were first incubated with 750 nM p37^AUF1^ on ice for 10 min, and then added with the RNA probe to incubate for 10 min.

### 4.3. Cell Culture

Human lung adenocarcinoma cells A549, human melanoma cells SK-MEL-2, human colon carcinoma cells HCT-116, human cervical cancer cells Hela, human breast cancer cells MCF-7, and human hepatocellular carcinoma cells HepG2 (American Type Culture Collection, Manassas, VA, USA) were grown in Dulbecco’s modified Eagle’s medium (GIBCO, Waltham, MA, USA) supplemented with 10% fetal bovine serum (Invitrogen, Waltham, MA, USA) at 37 °C in a humidified atmosphere containing 5% CO_2_ in air.

### 4.4. Cellular Thermal Shift Assay (CETSA)

SK-MEL-2, HCT-116, HepG2 cells were seeded on 10 mm cell dishes, and 3.5 μM of JNJ-7706621 (DMSO as negative control) was added when cell’s density reached 90% for one-hour incubation. Cells were then collected and heated at different temperatures followed by gradient cooling using a BIO-RAD T100 Thermal Cycler. Following repeated freeze–thaw cycles by liquid nitrogen to lyse cells, samples were centrifugated in 12,000 g rpm and liquid supernatant was determined by Western blot analysis. Image J software was used for densitometric analysis of protein expression. Melting curve was processed by GraphPad Prism 8.0 software (GraphPad Software, San Diego, CA, USA), using a sigmoidal model equation.

### 4.5. Real-Time Quantitative Reverse Transcription-PCR (qRT-PCR)

Cells were harvested and the total RNA was extracted by using the RNAiso Plus^®^ kit (Takara, Daliang, China). Total RNA was used as a template for reverse transcription using Reverse Transcriptase M-MLV (Takara, Daliang, China) according to the manufacturer’s instructions. The qPCR was done with primers (Appendix A) by using the THUNDERBIRD^®^ SYBR^®^ qPCR Mix (TOYOBO, Osaka, Japan) and carried out on a LightCycler480^®^ machine (Roche, Basel, Switzerland) as the followed cycle conditions: 95 °C denaturation (5 s, first cycle of 30 s), 60 °C annealing and extension (30 s), total 50 cycles. The relative expression level of a certain gene was calculated following the 2^−DDCt^ method and generate a column by using the GraphPad Prism 8.0 software (GraphPad Software, San Diego, CA, USA). To evaluate the remaining mRNA of target genes at different points, SK-MEL-2 cells were treated with 1.75 μM JNJ-7706621 for 24 h. Then cells were treated with 5 μg/mL Actinomycin D (Abmole, Shanghai, China) for 0, 3, 6, 9, and 11 h, respectively, to stop the transcription. The RNA was extracted and subjected to qRT-PCR with the above steps.

### 4.6. Western Blot

Cells were collected from 6-well plates and lysed using a RIPA lysis buffer. The whole-cell lysates were separated on an SDS-PAGE gel and transferred to nitrocellulose membranes. After blocking in 5% BSA, the membranes were incubated with primary antibodies and secondary antibodies as below: the anti-AUF1 antibody (CST, #D604F), the anti-IL8 antibody (Affinity Biosciences, #AF6342), the anti-GAPDH antibody (CST, #2118), and the HRP-conjugated anti-IgG antibody (Abcbm, #ab6728). The membranes were detected by ECL Plus kit (New Cell & Molecular Biotech Co., Ltd., Suzhou, China). The protein bands were visualized using chemiluminescence substrate.

### 4.7. RNA Interference

A validated siRNA duplex corresponding to the coding region of AUF1, and a negative control duplex were purchased from IGEbio Company (Guangzhou, China). The sequences of siAUF1 were forward-5′-GGTTATGGGAAGGTATCCATT-3′, reverse-5′-TGGATACCTTCCCATAACCdTdT-3′. For RNA interference experiment, SK-MEL-2 (1 × 10^5^ cells/well) cells were seeded in six-well plates allowing to grow to 30% confluency and transfected with 80 nM siRNA of AUF1 or a negative control according to the manufacturer’s instructions for 48 h. Cells were then collected by Trizol reagent (Accurate Biology, Hunan, China) as described above for further qRT-PCR or Western blot assays.

### 4.8. Overexpression of AUF1

The p37^AUF1^ gene sequence was inserted between *BamH* I and *EcoR* I sites of pcDNA3.1(+) vector. The plasmid was presented in the Match-T1 strain in the form of glycerobacteria (containing 10% glycerol final concentration). The glycerobacteria was inoculated into fresh LB medium (Amp, 1:1000 dilution) at 37 °C at 220 rpm for 18 h, and plasmid extraction was carried out by Plasmid small amount preparation Kit (GENEray, Shanghai, China). SK-MEL-2 (3 × 10^5^ cells/well) cells were seeded in six-well plates and transfected with 0.25 μg plasmid according to the manufacturer’s instructions for 24 h. Cells were then collected by Trizol reagent (Accurate Biology, Hunan, China) as described above for further qRT-PCR or Western blot assays.

### 4.9. Luciferase Reporter Detection

Cells were transfected with 100 ng reporter plasmids and 100 ng of pGL4.13 (Promega) expression plasmid in triplicates by using Lipofectamine 3000 (Invitrogen, CA, USA). After transfection for 48 h, cells were lysed and analyzed using the Dual-Luciferase Reporter Assay System (Promega) according to the manufacturer’s instruction. The ratio of Renilla Luciferase (*RLuc*) to Firefly Luciferase (*Luc*) expression was calculated to generate a column by using the Graphpad Prism 8.0 software.

### 4.10. RNA-Immunoprecipitation (RIP) Sequencing

Antibody immunoprecipitation RNA was quality control with Qubit™ (Thermo Fisher Scientifc, Waltham, MA, USA) and Agilent 2200 TapeStation (Agilent Technologies, Santa Clara, CA, USA). Then, 100 ng RNA was used for library building following NEBNext^®^ Ultra™ RNA Library Prep Kit protocol for Illumina (NEB, Ipswich, MA, USA). The final library product was assessed with Agilent 2200 TapeStation and Qubit^®^2.0 (Life Technologies, USA) and then sequenced on Illumina platformat sing-end/pair end reads for 50bp/150bp. Adaptor and low-quality bases were trimmed with Trimmomatictools (Version:0.36), and the clean reads undergone rRNA deleting through RNAcentralto get effective reads. Genomic alignment (version from UCSC genome browser) was using Tophat (Version:2.0.13) to get uniquely mapping reads. Effective reads form input sample can be used for RNA-seq analysis, the reads count value of each transcript was calculated by HTSeq (Version:0.6.0), then estimating RPKM (Reads Per Kilobase of transcript per Million mapped reads) value. RIP-seq data analyses were performed by RIBOBIO, Guangzhou, China.

### 4.11. RNA Sequencing

RNA-Seq analysis using purified total RNAs of SK-MEL-2 cells treated with 1.75 μM JNJ-7706621 were extracted using Trizol reagent and sent to PGEM Biotechnology Co., Ltd. (Guangzhou, China) for RNA-Seq analysis using an Illumina NovaSeq 6000 system after transcription. The quantity of gene expression was calculated by FPKM (Fragments Per Kilobase of transcript per Million fragments mapped). After quality control, the differentially expressed genes were defined with fold change ≥ 2 and *p* value ≤ 0.05. Volcano plots, Gene Cluster analyses, and enriched KEGG (Kyoto Encyclopedia of Genes and Genomes) pathway analyses were performed based on DEGs.

### 4.12. Statistical Analysis

All data are expressed as the mean ± SEM. Statistical comparisons were conducted using a *t* test or one-way analysis of variance (ANOVA) using the GraphPad Prism 8.0 software (GraphPad Software, San Diego, CA, USA).

## 5. Conclusions

In the present study, a small-molecule compound, JNJ-7706621, was found to bind AUF1 protein and destroy the homeostasis of mRNA degradation by screening the commercial compound library in the present study. JNJ-7706621 bound to AUF1 and intervened in forming the AUF1-RNA complex, leading to reduced mRNA degradation. Based on the AUF1 target gene IL8, an essential pro-inflammatory factor, we found that the RNA-protein interaction intervention inhibited gene expression in the context of destroying mRNA homeostatic state, which might be associated with intracellular accumulation of IL8 mRNA. As a proof-of-concept experiment, we designed and synthesized three structural analogs based on JNJ-7706621 and found that analogs **1** and **3** had an inhibitory effect on RNA-AUF1 interaction. These studies provide innovative drug design strategies to regulate mRNA homeostasis.

## Figures and Tables

**Figure 1 molecules-27-03182-f001:**
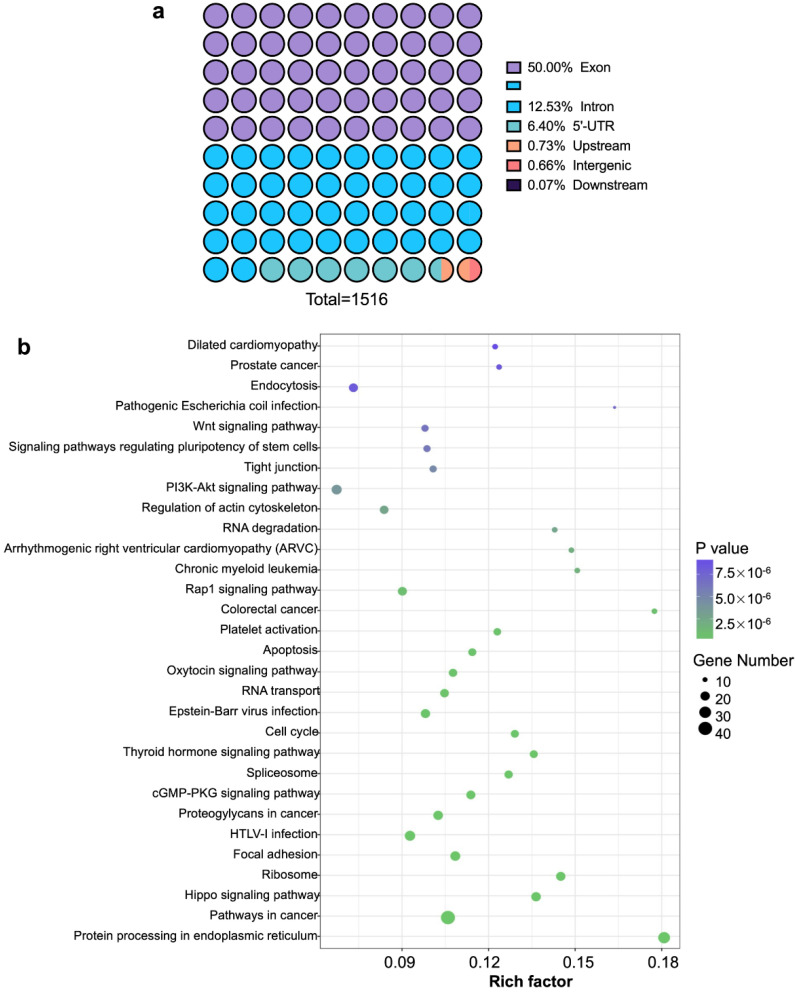
RIP-seq analyses of detecting AUF1’s function in Hela cells. (**a**) Statistics of AUF’s nuclear localization annotation results. Binding peaks were annotated by Homer software, and annotation results were counted. (**b**) KEGG pathway enrichment of AUF1 target genes. The *X*-axis shows the Rich Factor (the ratio of the number of target genes enriched in this pathway to the number of annotated genes), and the left *Y*-axis shows pathways’ names of target genes. The range of *p*-value is (0,1]. The experiments were parallelly repeated in twice.

**Figure 2 molecules-27-03182-f002:**
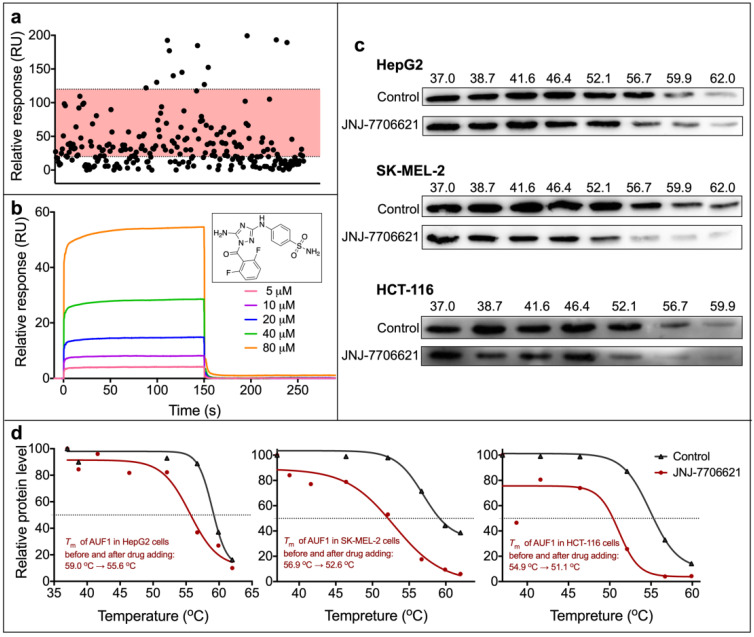
Screening of AUF1’s binder by SPR assay and the target validation. (**a**) Pre-screening from the commercially available compounds library by SPR assay. The concentration of compounds was 50 μM. Compounds with a RU value ranging from 20 to 120 (red region) were selected for further kinetic analysis. (**b**) The sensorgram and structure of compound JNJ-7706621 in the kinetic analysis. (**c**) Results of the cellular thermal shift assay (CETSA) of AUF1 in three cell lines. Cells were treated with JNJ-7706621 at 1, 3.5, and 6 μM, respectively. From upper to bottom: Western blot graph in HepG2, SK-MEL-2, and HCT-116 cells, respectively. (**d**) Melting curves from the CETSA experiments. The curves were obtained by GraphPad Prism 8.0 and fitted with a sigmodal mode to get the melting temperatures. From left to right: curves and melting temperatures (*T*_m_) in HepG2, SK-MEL-2, and HCT-116 cells.

**Figure 3 molecules-27-03182-f003:**
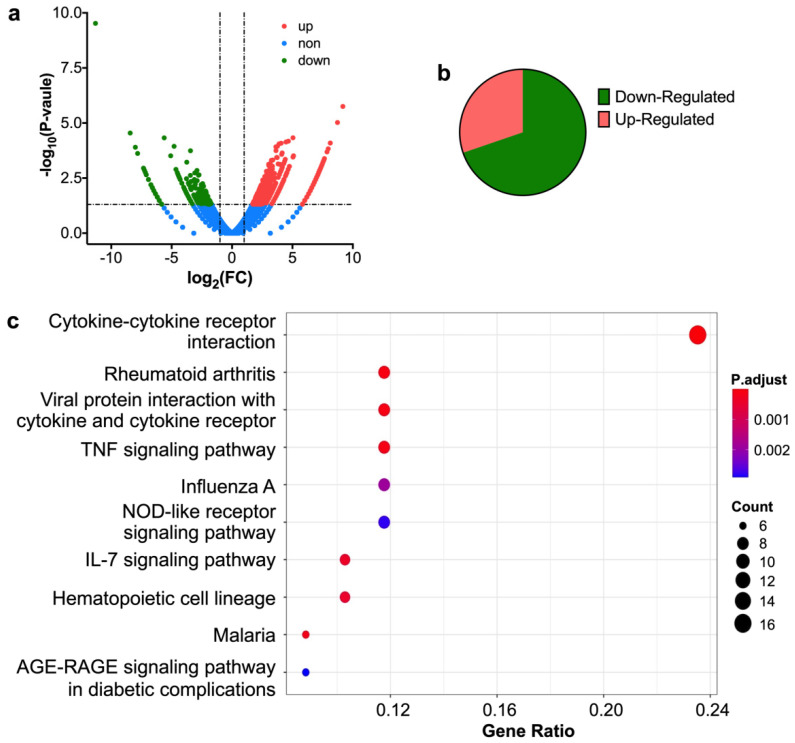
RNA-seq analysis of JNJ-7706621 on the gene expression profile in SK-MEL-2 cells. (**a**) The volcano plots visualize differentially expressed genes (DEGs) between the JNJ-7706621-treated and the control group. The *p* < 0.05 was used as the significance threshold to determine the significance of DEGs. (**b**) The percentage of up-regulated and down-regulated genes in DEGs. (**c**) KEGG pathway enrichment of JNJ-7706621-regulated genes. The range of *p*-value is (0,1). The experiments were parallelly repeated in twice.

**Figure 4 molecules-27-03182-f004:**
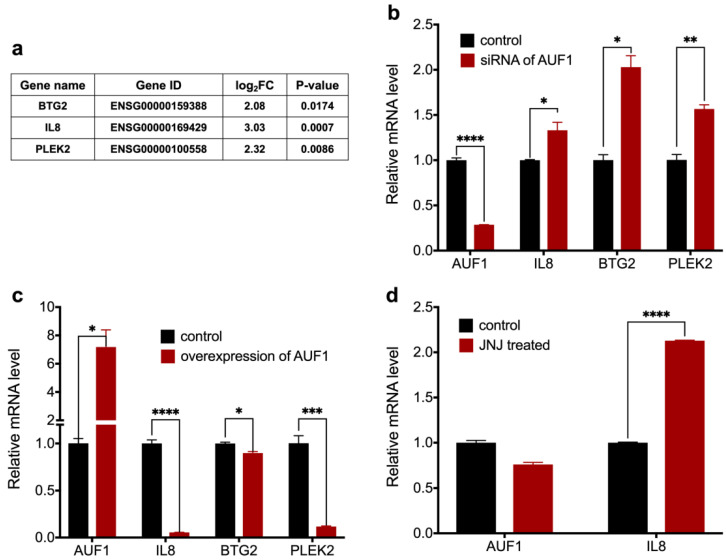
Identification of AUF-1e fand JNJ-7706621on of AUF-1e for (**a**) Differentially expressed genes with the most significant difference combining the data in RNA-seq and RIP-seq. (**b**) The relative mRNA level of IL8, BTG2, and PLEK2 after knocking down the AUF1 gene using siRNA. The siRNA of AUF1 (80 nM) was transfected to SK-MEL-2 cells for 48 h. (**c**) The relative mRNA level of IL8, BTG2, and PLEK2 after overexpressing the AUF1 gene. The plasmid containing the p37AUF1 sequence (0.25 μg) was transfected to SK-MEL-2 cells for 24 h. (**d**) The relative mRNA level of IL8 after treatment with JNJ-7706621 at 1.75 μM for 24 h. Data are shown as means ± SEM of *n* = 3. * *p* < 0.05, ** *p* < 0.01, *** *p* < 0.005, **** *p* < 0.001. All experiments were parallelly repeated in triplicate.

**Figure 5 molecules-27-03182-f005:**
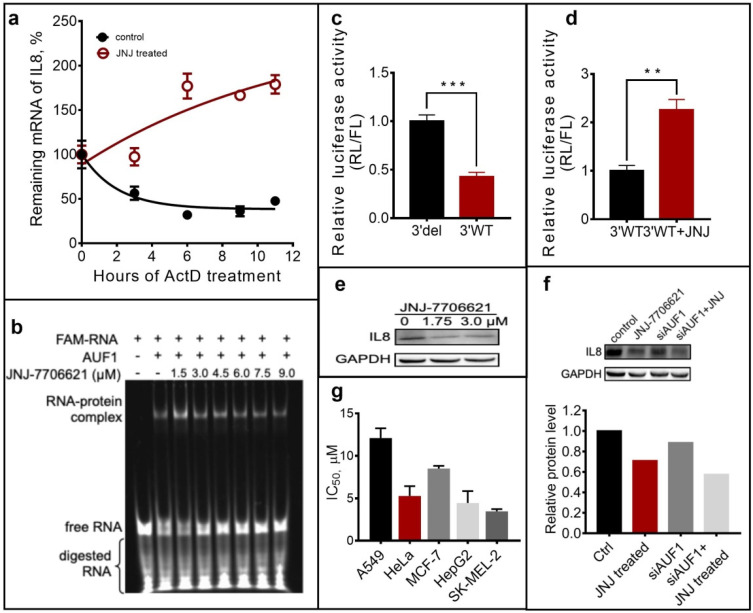
JNJ-7706621 regulated IL-8 gene’s translation via interacting with AUF1 and inhibited tumor cells’ proliferation. (**a**) The remaining amount of *IL8* mRNA after treated with 1.75 μM JNJ-7706621 for 24 h in SK-MEL-2 cells. Actinomycin D (5 μg/mL) was added to stop the transcription to detect remaining *IL8* mRNA with qRT-PCR. The fitting curve was performed using GraphPad Prism. (**b**) The REMSA result of AUF1 (750 nM) incubated with the FAM-labelled RNA at 250 nM in the presence of JNJ-7706621. The sample was incubated in an incubation buffer (10 mM Tris-HCl, pH 8.0, containing 50 mM KCl, 2 mM DTT, 0.5 mM EDTA, and 0.1 µg/µL BSA) and loaded on a 6% native polyacrylamide gel. (**c**) The wild-type plasmid (3′WT) and mutant plasmid (3′del) were transfected in SK-MEL-2 cells. After incubation, the relative luciferase activity (RL/FL) was detected and normalized with the data of 3′del. (**d**) The relative luciferase activity of 3′WT and 3′WT dual-luciferase reporters after treated with JNJ-7706621 (1.75 μM), normalizing with the data of 3′WT. (**e**) The expression of IL8 in SK-MEL-2 cells treated with 1.75 and 3 μM JNJ-7706621. (**f**) The expression of IL8 in SK-MEL-2 cells treated with 80 nM siRNA of AUF1. JNJ-7706621 was added after 10-h pre-incubation with the siRNA. (**g**) Inhibitory effect of JNJ-7706621 in different cancer cells for 48 h in the MTT assay. All experiments were parallelly repeated in triplicate, and data are shown as means ± SEM of *n* = 3. ** *p* < 0.01, **** *p* < 0.001.

**Figure 6 molecules-27-03182-f006:**
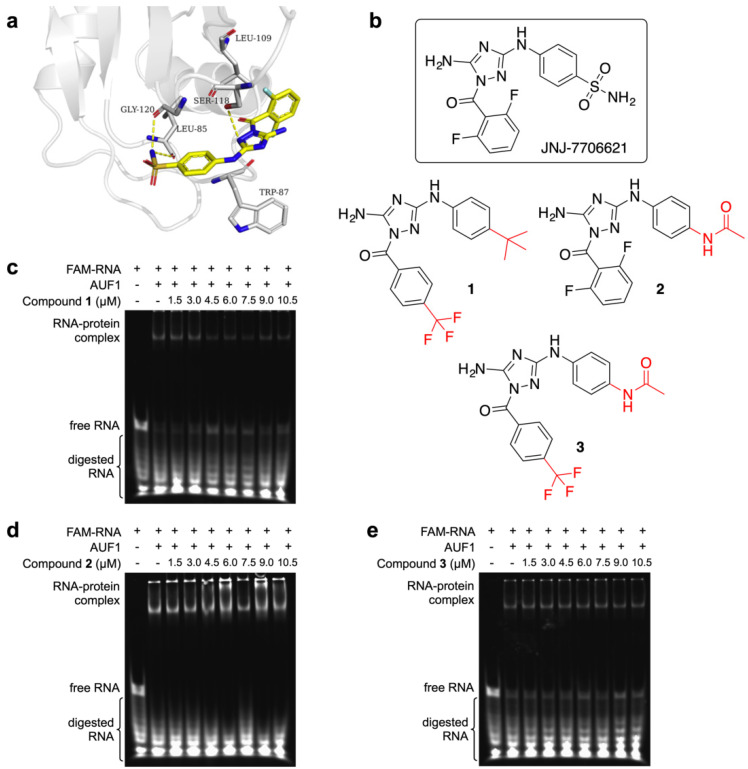
Proof-of-concept design based on the molecular docking. (**a**) Preferred docking pose of JNJ-7706621 (depicted in yellow) bound to the AUF1. (**b**) Comparison of new-synthesized compound 1, 2, and 3 with JNJ-7706621. (**c**–**e**) The REMSA result of AUF1 (750 nM) incubated with the FAM-labelled RNA at 250 nM in the presence of compound **1**, **2**, and **3**.

## Data Availability

Not applicable.

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
