# Peer review of "A Novel Strategy for Regulating mRNA’s Degradation via Interfering the AUF1’s Binding to mRNA"

_molecules, 2022, doi:10.3390/molecules27103182_

Round 1
Reviewer 1 Report
The authors screened a drug library to identify AUF1-interactors. They found that JNJ-7706621 binds to AUF1, which is expected to affect mRNA degradation. The formulation of the text is often unclear and incomplete that is very difficult to assess validity of findings. Examples are indicated below.
Major comments
- The number of biological replicate experiments should be stated for each experiment, preferentially in each figure caption. This is particularly important in plots where P-values are displayed (e.g. Fig. 3a).
- 5a. This figure is very important for the overall message of the manuscript but there are several critical points. The two colors are not labelled. The experimental detail is not described: It is not clear whether JNJ-7706621 was added at the same time as Actinomycin D. Importantly, the brown data display an increase in RNA levels-which means that half-life cannot be calculated unless the production rate is characterized. Alternatively, ActD should be added after a 1 day pre-incubation with JNJ-7706621.
- Chapter 2.5. The title is misleading: “Novel JNJ-7706621’s analogs showed similar effects on mRNA’s stability”. However, mRNA half-life (stability) is not measured in this section at all. Only the REMSA is performed, which is an mobility shift assay, which does not reveal mRNA half-lives.
- Misleading terms: “self-degradation of mRNA”. Do the authors really believe that an mRNA degrades itself? If yes, they should cite all the relevant literature.
Minor comments
- RNA-seq analysis (Fig. 3). In addition to the enrichment of functions, the 5 to 10 most strongly affected genes should be stated explicitly.
- There are many formulations that are unclear. The English of the text should be improved. Examples:
“The high distribution”
Is the Rich-factor a proper noun or it is simply the enrichment factor?
“to freeze the transcription for detection” For the detection of what?”
Reviewer 2 Report
Comments to the Author
In the manuscript by Li KT. et al., the authors describe a novel strategy for interfering with AUF1-RNA binding by employing a small-molecule compound, JNJ-7706621. The authors shows that JNJ-7706621 could inhibit IL8 expression by destroying the mRNA homeostatic state. Overall, this manuscript constitutes conceptual framework for understanding recent advances in the field of AUF1-regulated gene expression, however, detailed descriptions regarding central topics to this theme are missing in some places.
Major points
1, In Figure 2C and 2D, the authors performed the cellular thermal shift assay and show that JNJ-7706621 interacts with AUF1 and reduces the protein stability. However, it is necessary to show whether JNJ-7706621 mediates the stability under physiological conditions. To reveal the effect to AUF1 protein stability, it is reasonable to show the time course change of AUF1 protein expression in cells supplemented with cycloheximide in addition to JNJ-7706621 using western blot.
2, Regarding the results shown in Figure 4D, the author’s interpretation of the results is ambiguous. In order to investigate whether AUF1 is involved in up-regulation of IL-8 expression in cells treated with JNJ-7706621, AUF1 knockdown experiments are required.
3, In Figure 5D and 5E, the authors show that expression levels of reporter mRNA containing IL-8 3’UTR were increased (Fig 5D), but IL-8 protein expression was decreased by administration of JNJ-7706621. The authors mention that negative feedback or destroying mRNA homeostatic state caused the discordant expression, however, the claim is not convincing. Considering that in general 3’UTR contains various cis-elements, including microRNA recognition element, it is possible that other trans-acting factors, such as microRNA, are more likely to interact with IL-8 mRNA in the absence of AUF1. The authors need to mention the possibility of the participation of other trans-acting factors in IL-8 expression by referring previous studies, for example;
AUF1, an mRNA decay factor, has a discordant role in Cpeb1 expression
Oe S, Koike T, Hirahara Y, Tanaka S, Hayashi S, Nakano Y, Kase M, Noda Y, Yamada H, Kitada M.
Biochem Biophys Res Commun. 2021 Jan 1;534:491-497.
doi: 10.1016/j.bbrc.2020.11.054. Epub 2020 Nov 19.
Cpeb1 expression is post-transcriptionally regulated by AUF1, CPEB1, and microRNAs.
Oe S, Hayashi S, Tanaka S, Koike T, Hirahara Y, Kakizaki R, Sakamoto S, Noda Y, Yamada H, Kitada M.
FEBS Open Bio 2022 022 Jan;12(1):82-94.
doi: 10.1002/2211-5463.13286. Epub 2021 Nov 8.
MicroRNA miR-93-5p regulates expression of IL-8 and VEGF in neuroblastoma SK-N-AS cells.
Fabbri E, Montagner G, Bianchi N, Finotti A, Borgatti M, Lampronti I, Cabrini G, Gambari R. Oncol Rep. 2016 May;35(5):2866-72.
doi: 10.3892/or.2016.4676. Epub 2016 Mar 15.
Minor points
The authors use the word ‘knock-in’ in experiments using transient overexpression, for example lines 165 and 167. In general, knock-in means a genetic engineering technique to insert a complementary DNA sequence encoding a protein at a specific locus on a chromosome, therefore, the authors should correct the text.
Round 2
Reviewer 1 Report
The authors addressed my comments 1, 3, 4, 5 and 6. However, comment 2 was not addressed. If a transcription inhibitor is added, the RNA level must decline and not increase unless the results represent an artifact, which seems to be the case. Given the artifact, the mRNA half-life cannot be measured and no fitting makes sense with this technique. If the authors want to measure the half-life, they can use metabolic labelling, and they can determine both the decay (half-life) and production rates.
Author Response
Thanks for your advice. We realized that the half-life value is not appropriate at this situation. So, we have deleted the statement and calculation of "half-life" and used "remaining mRNA" instead. The related statement is as follow: “The mRNA amount decreased with time in the control group while increased in the JNJ-7706621 treated group. Compared with the control group, mRNA levels were higher at each time point after treated with JNJ-7706621, indicating an inhibition effect on mRNA decay.”Reviewer 2 Report
The authors made appropriate revisions to the issues we pointed out.
Author Response
Thanks for your efforts.